

# Prediction of PM$_{2.5}$ concentration based on a CNN-LSTM neural network algorithm

Xuesong Bai[1], Na Zhang[1], Xiaoyi Cao[2] and Wenqian Chen[1]

[1] School of Information and Control Engineering, Qingdao University of Technology, Qingdao, Qingdao City, Shandong Province, China
[2] Key Laboratory for Semi-Arid Climate Change of the Ministry of Education, College of Atmospheric Sciences, Lanzhou University, Lanzhou, Lanzhou City, Gansu Province, China

## ABSTRACT

Fine particulate matter (PM$_{2.5}$) is a major air pollutant affecting human survival, development and health. By predicting the spatial distribution concentration of PM$_{2.5}$, pollutant sources can be better traced, allowing measures to protect human health to be implemented. Thus, the purpose of this study is to predict and analyze the PM$_{2.5}$ concentration of stations based on the integrated deep learning of a convolutional neural network long short-term memory (CNN-LSTM) model. To solve the complexity and nonlinear characteristics of PM$_{2.5}$ time series data problems, we adopted the CNN-LSTM deep learning model. We collected the PM$_{2.5}$ data of Qingdao in 2020 as well as meteorological factors such as temperature, wind speed and air pressure for pre-processing and characteristic analysis. Then, the CNN-LSTM deep learning model was integrated to capture the temporal and spatial features and trends in the data. The CNN layer was used to extract spatial features, while the LSTM layer was used to learn time dependencies. Through comparative experiments and model evaluation, we found that the CNN-LSTM model can achieve excellent PM$_{2.5}$ prediction performance. The results show that the coefficient of determination (R$^2$) is 0.91, and the root mean square error (RMSE) is 8.216 µg/m$^3$. The CNN-LSTM model achieves better prediction accuracy and generalizability compared with those of the CNN and LSTM models (R$^2$ values of 0.85 and 0.83, respectively, and RMSE values of 11.356 and 14.367, respectively). Finally, we analyzed and explained the predicted results. We also found that some meteorological factors (such as air temperature, pressure, and wind speed) have significant effects on the PM$_{2.5}$ concentration at ground stations in Qingdao. In summary, by using deep learning methods, we obtained better prediction performance and revealed the association between PM$_{2.5}$ concentration and meteorological factors. These findings are of great significance for improving the quality of the atmospheric environment and protecting public health.

## INTRODUCTION

Air pollution has become an important environmental problem, among which fine particulate matter (PM$_{2.5}$) is widely regarded as one of the pollutants that has the

Corresponding author
Wenqian Chen, chimmyqu@yeah.net

greatest impact on human health and the ecological environment (*Kioumourtzoglou et al., 2016*). Therefore, accurate prediction of $PM_{2.5}$ concentration is of great significance for environmental monitoring, pollution control and public health protection (*Battye, Aneja & Roelle, 2003*). The traditional statistical method and the simple machine learning prediction model have some limitations, making it difficult to address the complex nonlinear relationships and spatiotemporal variation characteristics of $PM_{2.5}$ at ground stations (*Yu et al., 2020*). Therefore, how to effectively capture spatial and temporal characteristics, address time series dependence and develop an intelligent estimation method for predicting $PM_{2.5}$ at ground stations have become technical hot spots (*Xiao et al., 2017*).

In studies on $PM_{2.5}$ prediction methods, some statistical methods, such as the autoregressive comprehensive moving average (ARIMA) (*Wang & Wang, 2023*) model, have been introduced. While these methods have been used to predict air quality, studies have shown that if the temporal and spatial series of $PM_{2.5}$ are nonlinear, some linear statistical models cannot be applied. Therefore, some scholars have used machine learning methods such as the support vector regression (SVR) (*Armaghani et al., 2020*) model to predict $PM_{2.5}$ at stations. These methods have been used in various regional predictions, but there are still certain limitations. For example, a SVR model with implicit kernel mapping (such as a radial basis function (RBF) kernel) may not be able to obtain an air quality prediction model with good performance because air quality predictors use a large amount of complex data, which can lead to model overfitting phenomena. In recent years, to overcome the shortcomings of such models, artificial neural networks (ANNs) in machine learning have been widely used in the prediction of $PM_{2.5}$ and have achieved good results. *Fu (2016)* used hour-by-hour data from the monitoring station in Baoji City, Shaanxi Province, selected six air pollutants as factors influencing $PM_{2.5}$ concentration, used a fuzzy neural network and a backpropagation (BP) neural network optimized by a genetic algorithm to predict $PM_{2.5}$ concentration. *Mohamed (2019)* used an artificial neural network to predict the air quality index in Ahvaz, Iran, and proved its applicability through a comparative test. *Zhang et al. (2022)* predicted the average daily air pollution index (API) of Shanghai based on a multilayer BP neural network and selected and optimized the hidden layer nodes of the model through repeated experiments, effectively improving the prediction accuracy of the model. *Yin et al. (2016)* used a multivariate nonlinear regression model to predict the daily $PM_{2.5}$ concentration in Beijing by using coarse particulate matter ($PM_{10}$) andwind speed and direction as independent variables. In the application of the BP neural network, *Shi et al. (2005)* conducted a sensitivity analysis on the training frequency of the model and found that increasing the training frequency was helpful for improving the model performance. However, if the training frequency continues to increase, the model exhibits an overfitting phenomenon, reducing the accuracy of the model.

However, due to the rapid increase in the amount and dimension of data used in $PM_{2.5}$ prediction models, simple machine learning no longer meet the model requirements. As a new artificial intelligence technology, deep learning has achieved remarkable results in different fields, such as computer vision, text processing and time series prediction (*Xu et al., 2023*). For example, deep neural networks have recently been used to predict

air pollutants and have performed well. Some scholars have used the long short-term memory (LSTM) model to predict air pollutants in various regions of the world (*Liu et al., 2020*). Due to the excellent performance of this model in time series problems, it can predict air pollutants well. At the same time, there are still shortcomings that need to be improved, such as the inability of a single LSTM model to learn spatial information. Specifically, the concentration of air pollutants changes with their emission, diffusion and reaction with other suspended particles, indicating that air pollutants have extremely important correlations with spatial dimensions and should not be ignored (*Kim et al., 2022*). In addition, other scholars have used a convolutional neural network (CNN) for the prediction of $PM_{2.5}$. It has been shown that the spatial dimension has a strong processing capacity, and the solution to air pollution prediction in spatial correlation is a reasonable method. This approach is widely used in image recognition, but at the same time, it has some shortcomings in terms the prediction of time performance. Therefore, it can be seen from the above literature that $PM_{2.5}$ prediction has now stepped into the field of deep learning, but some deep learning methods cannot achieve the time and space information extraction of $PM_{2.5}$ (*Li, Liu & Zhao, 2022*). Therefore, how to better combine deep learning with the temporal and spatial dimensions of $PM_{2.5}$, accurately predict its development trend, and effectively increase the explanatory degree in terms of the data volume and dimension of the model are the current technical bottlenecks faced in air pollutant prediction for most regions (*Eren, I & Erden, 2023*).

Based on the above research, we proposed a method for predicting $PM_{2.5}$ concentrations at stations based on a convolutional neural network long short-term memory (CNN-LSTM) neural network. Traditional prediction methods are often unable to address complex nonlinear relations and spatiotemporal changes. In this article, a CNN and an LSTM network are combined to effectively extract spatiotemporal features and learn time series dependencies, thereby improving the prediction accuracy. This method can address the temporal and spatial correlations between sites. $PM_{2.5}$ concentration changes at a site are usually affected by the neighboring sites. The proposed method can consider the relationships between multiple sites at the same time, further improving the accuracy and global prediction results. This feature is of great significance in urban air quality and ecological environment monitoring and improvement. In summary, the main contribution of this article is the proposal of a method for predicting the $PM_{2.5}$ concentrations at sites based on a CNN-LSTM neural network, which effectively extracts the temporal and spatial feature dependencies, learns the time series, processes the spatial correlations between sites, and effectively improves the accuracy of regional $PM_{2.5}$ prediction by optimizing the model structure and algorithm. This research has important theoretical and technical value for environmental monitoring and protection, public health and decision-making.

# DATA

## Study area

We chose Qingdao, an important city in China, as the research area. Qingdao is located in the south of Shandong Peninsula, east of the Yellow Sea. In Qingdao, the population has

grown rapidly in recent years. While many enterprises have flourished, this region is facing challenges related to the rapid development of economic, industrial and urbanization processes. Human factors have led to a sharp increase in the emission of air pollutants, such as $PM_{2.5}$, and unique giant sea salt aerosol particles combine with abundant urban aerosols, resulting in a secondary aerosol effect (light pollution problem). Air pollution has become one of the factors restricting sustainable development. To understand the $PM_{2.5}$ concentration in Qingdao, a study was conducted based on 20 national ground monitoring stations for $PM_{2.5}$. The study area is shown in Fig. 1.

## Data sources

Daily historical air quality concentration and meteorological data of Qingdao city from January 1, 2020, to December 31, 2020, were chosen as the research objects and used as input parameters of the CNN-LSTM model proposed in this study. The main air pollutant we chose was $PM_{2.5}$. Temperature, pressure, and wind speed were selected as meteorological data indicators. $PM_{2.5}$ data were obtained from the Qingdao station data, and the meteorological data used in this study were retrieved from the China Meteorological Data Service Center (http://data.cma.cn/en). Data were interpolated by Kriging interpolation method in ArcGIS software to obtain continuous coverage of meteorological data in the study area, and finally a surface meteorological data set of Qingdao with a resolution of 1km in 2020 was obtained. $PM_{2.5}$ imposes a strong random effect in space and time. Thus, we selected three meteorological factors that have the greatest influence on $PM_{2.5}$ concentration over Qingdao, namely temperature (TEM), wind speed (IWS) and pressure (PRS). Therefore, the proposed CNN-LSTM model can be evaluated from the data of 20 stations in Qingdao. Table 1 shows the names and information of the monitoring stations.

## Data normalization processing and data division

We found that after $PM_{2.5}$ data acquisition at the abovementioned sites, the obtained data are usually dirty, with missing values and inconsistent units (*Narkhede et al., 2023*). Therefore, it is not possible to use these data directly for the model training and relationship mining. To improve the model prediction accuracy and reduce the time needed for the actual training, we carried out pretreatment operations on the $PM_{2.5}$ and meteorological data (*Zheng et al., 2022*). The specific cleaning operations applied to this dataset include modifying the year, month and day information as an index, deleting the data with continuous null values, fill in the missing values with the value of the previous known data. This method is relatively simple and effective, which can maintain the continuity of data and help reduce information loss, thereby improving the accuracy of the model. standardize and normalize the dataset to ensure that it meets the input requirements of the neural network, and averaging the data for every 24 h and reporting the results in hours (*Liang et al., 2020*). In the model training, due to the different ranges of variable data, it is necessary to normalize the feature data. The normalization formula is shown in Eq. (1):

$$M = \frac{X - X_{min}}{X_{max} - X_{min}} \tag{1}$$

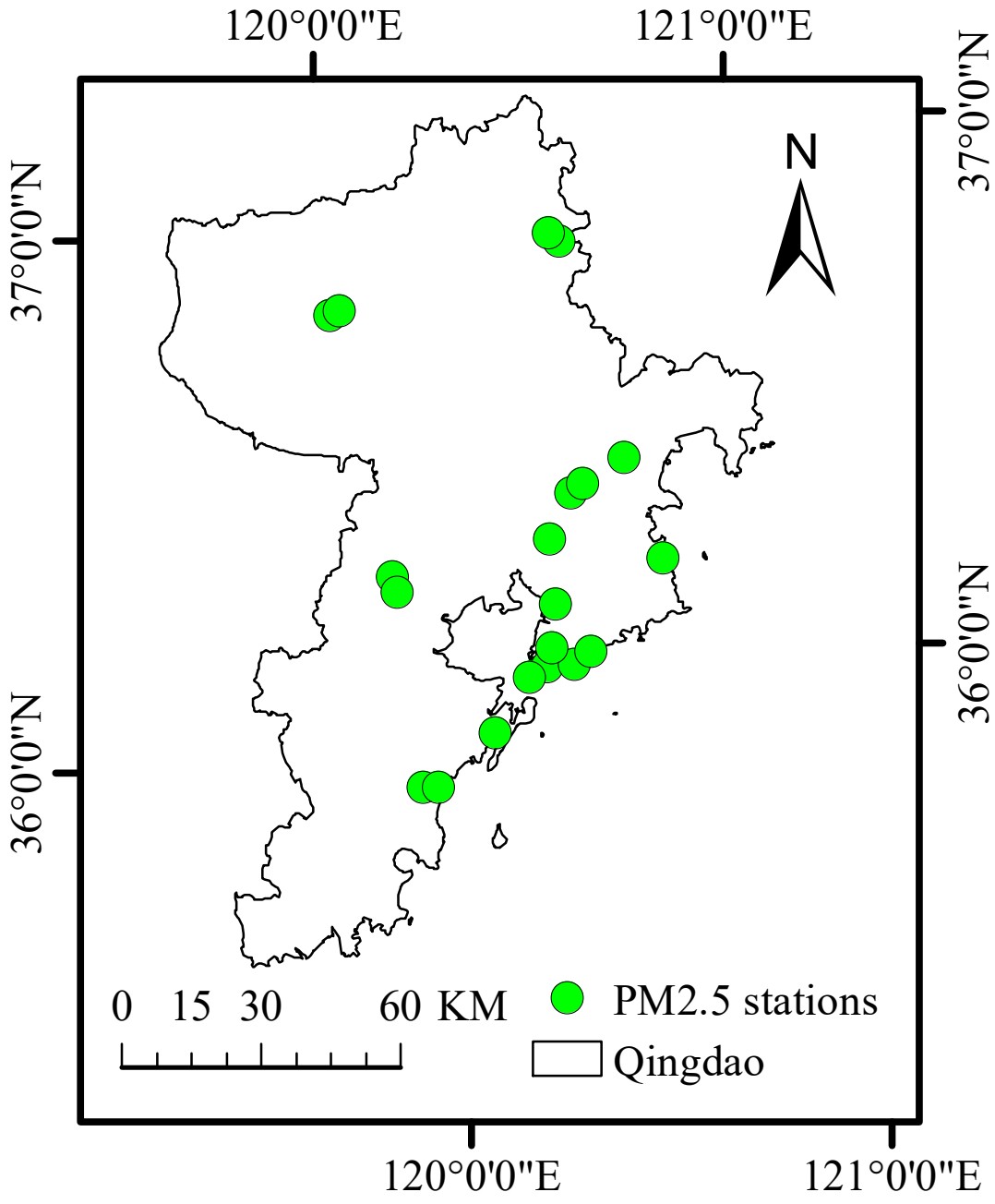

**Figure 1   Study area and PM$_{2.5}$ stations.** Map created using ArcMap.

where M is the value after x normalization, x is the original value of the variable, X$_{min}$ is the minimum value, and X$_{max}$ indicates the maximum value. After the original value of each variable is normalized, the dataset is divided into a training set (80%) and a test set (20%) for the input parameters of the model in this study to provide the data utilization efficiency and the model accuracy (*Sun & Xu, 2021*).

**Table 1  PM$_{2.5}$ site information.**

| Station | Longitude | Latitude | Station | Longitude | Latitude |
|---|---|---|---|---|---|
| Yangkou | 120.67 | 36.24 | Develop substation | 120.47 | 36.39 |
| Licang station | 120.39 | 36.19 | Jiaonan station | 120.00 | 35.88 |
| Shibei substation | 120.35 | 36.07 | Jiaozhou station | 120.01 | 36.28 |
| Shinan east station | 120.41 | 36.07 | Jiaozhou | 120.02 | 36.25 |
| Huangdao substation | 120.20 | 36.96 | Square substation | 120.52 | 36.89 |
| Sifang substation | 120.37 | 36.10 | Laoshan substation | 120.46 | 36.09 |
| Shinan west | 120.30 | 36.01 | Pingdu1 | 119.95 | 36.79 |
| Lacey | 120.54 | 36.87 | Pingdu2 | 119.98 | 36.80 |
| Chengyang substation | 120.40 | 36.31 | Jimo1 | 120.61 | 36.44 |
| Huangdao district | 120.04 | 35.87 | Jimo2 | 120.50 | 36.40 |

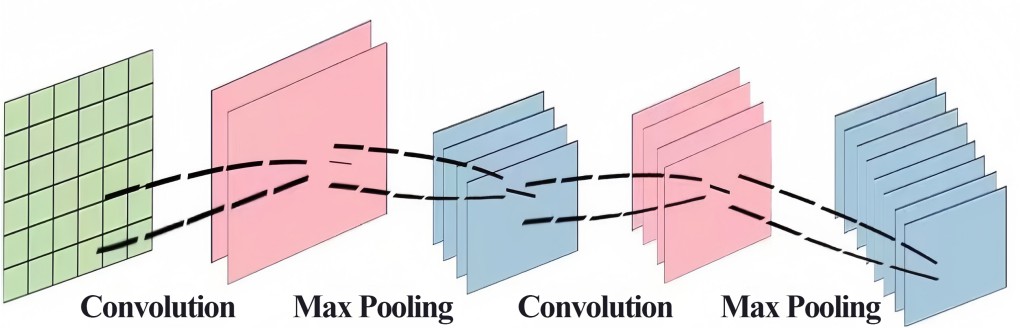

**Convolution     Max Pooling     Convolution     Max Pooling**

**Figure 2  CNN structure chart.**

## MODELS AND METHODS

### CNN

It was mentioned in the introduction of this article that CNNs are currently used in air pollutant prediction to help extract the spatial information of pollutants. A CNN is a kind of deep neural network that is generally composed of an input layer, a convolutional layer, a pooling layer, a fully connected layer and an output layer. The convolution and pooling layers are generally applied for feature engineering, and the fully connected layer is used for feature weighting and is equivalent to a ''classifier'' (*Chua, 1997*; *Chua & Roska, 1993*; *Girshick, 2015*). The network features include ''local connectivity'' and ''weight sharing'', which simplify the complexity of network links, improve the ability of the model to extract abstract features, and alleviate the problems of slow training and easy overfitting of the fully connected network to a certain extent (*Bhatt et al., 2021*). The convolutional neural network structure is shown in Fig. 2.

### LSTM

It was mentioned in the introduction that LSTM has great advantages in PM$_{2.5}$ time series prediction. First, recurrent neural networks (RNNs) are a class of recurrent neural networks
that use time series as input. The input condition of a recurrent neural network lead to its natural advantage in processing time series, and its parameter sharing structure enables it to better learn the nonlinear features of sequences (*Staudemeyer & Morris, 2019*; *Yu et al., 2019*). LSTM is an improved recurrent neural network model that realizes information screening and retention by introducing three "gate" structures: an input gate, an output gate and a forget gate (*Qin et al., 2022*). Figure 3 shows the internal structure of the LSTM unit, and Fig. 4 illustrate the LSTM network structure. $C_{t-1}$ represents the state of the cell at a previous time ($t-1$ time), $h_{t-1}$ represents the state of the hidden layer of the cell at a previous time ($t-1$ time), $x_t$ represents the input at the current time (T-time), $C_t$ is the state of the cell after an update (memory unit), $h_t$ is the state of the hidden layer after an update (memory unit), and HT is the input of the subsequent cell unit(*Song et al., 2020*). As mentioned earlier, the three gate structures within the cell unit, the forget gate $f_t$, the input gate $i_t$, and the output gate $O_t$, are responsible for updating the cell state and the hidden layer state. The specific functions of the three gates are as follows. The forget gate $f_t$ determines how much the cell state $C_{t-1}$ transitions from the previous moment ($t-1$ moment) to the current moment, the input gate $i_t$ determines how much input $x_t$ from the current moment (T-moment) contributes to the cell state $C_t$ from the current moment, and the output gate determines how much cell state $C_t$ from the current moment (T-moment) influences the output $h_t$. Unlike the traditional RNN, which directly outputs $h_{t-1}$ from the hidden layer at the previous time as a reflection of the historical state, the LSTM network uses the cell state $C_{t-1}$ at the previous time as a reflection of the historical state through the self-circulating memory unit inside the cell and constantly updates the hidden layer state of the network, which has been proven to exhibit better timing processing capability and applicability in practice (*Qadeer et al., 2020*). The corresponding cell renewal process is shown in Eqs. (2)–(7):

$$f_t = \sigma(W_f \cdot [h_{t-1}, x_t] + b_f) \tag{2}$$

$$i_t = \sigma(W_i \cdot [h_{t-1}, x_t] + b_i) \tag{3}$$

$$\overline{C}_t = tanh(W_c \cdot [h_{t-1}, x_t] + b_c) \tag{4}$$

$$C_t = f_t \times C_{t-1} + i_t \times \overline{C}_t \tag{5}$$

$$O_t = \sigma(W_o \cdot [h_{t-1}, x_t] + b_o) \tag{6}$$

$$h_t = O_t \times tanh(C_t) \tag{7}$$
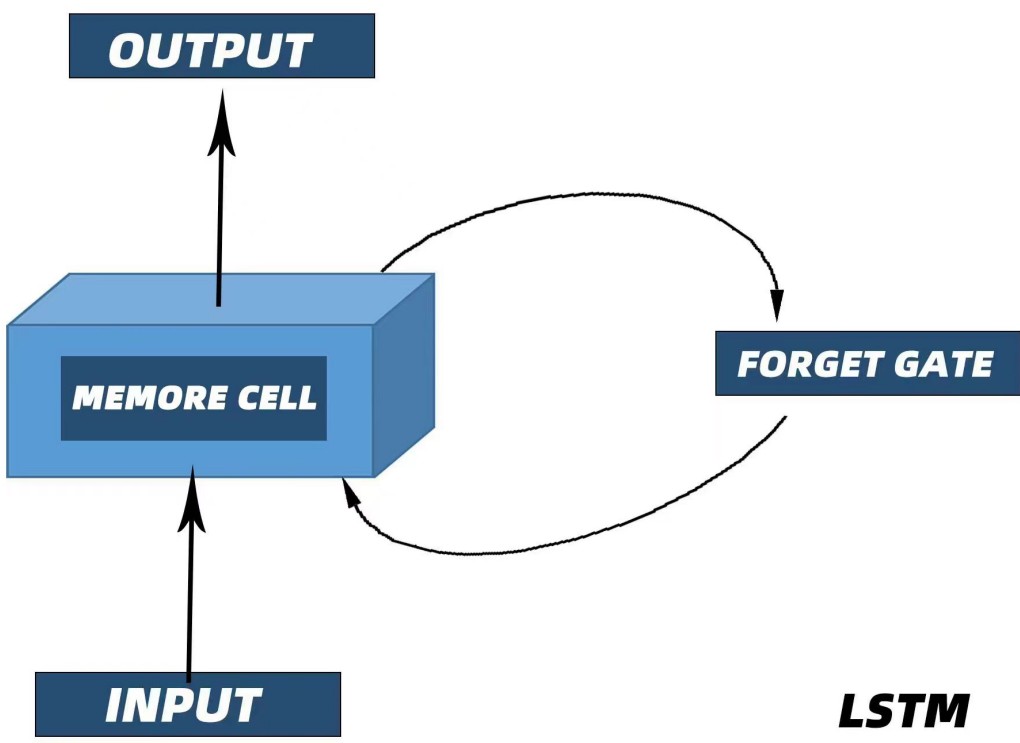

Figure 3 **Internal structure of the LSTM unit.**

In these formulas, $W_f$, $b_f$, $W_i$, $b_i$, $W_c$, $b_c$, $W_o$ and $b_o$ are the parameters of the LSTM model that are constantly updated during the training process. σ and tanh are the activation functions of the hidden layer of the model, which are responsible for improving the nonlinear representation of the model.

## Integrated CNN-LSTM neural network model

In this study, we combined the CNN and LSTM models to address the time series prediction problem and effectively improve the PM$_{2.5}$ estimation accuracy. We used the CNN to extract the features of the time series data and designed the LSTM according to the output of the CNN model for prediction (*Huang & Kuo, 2018*).

The specific structure of the CNN-LSTM model we constructed is shown in Fig. 5. We extracted PM$_{2.5}$, temperature, pressure and wind speed information from 20 stations in Qingdao, interpolated the missing values, and normalized the data. The processed data were input into the CNN. In considering the particularity of PM$_{2.5}$ time series data, a one-dimensional convolutional layer and pooling layer were designed as the base layer of the hybrid model (*Li, Hua & Wu, 2020*). In order to better input the output of CNN and LSTM to the fully connected layer, a flat layer is constructed between the LSTM layer and the fully connected layer. A fully connected layer is constructed to decode the
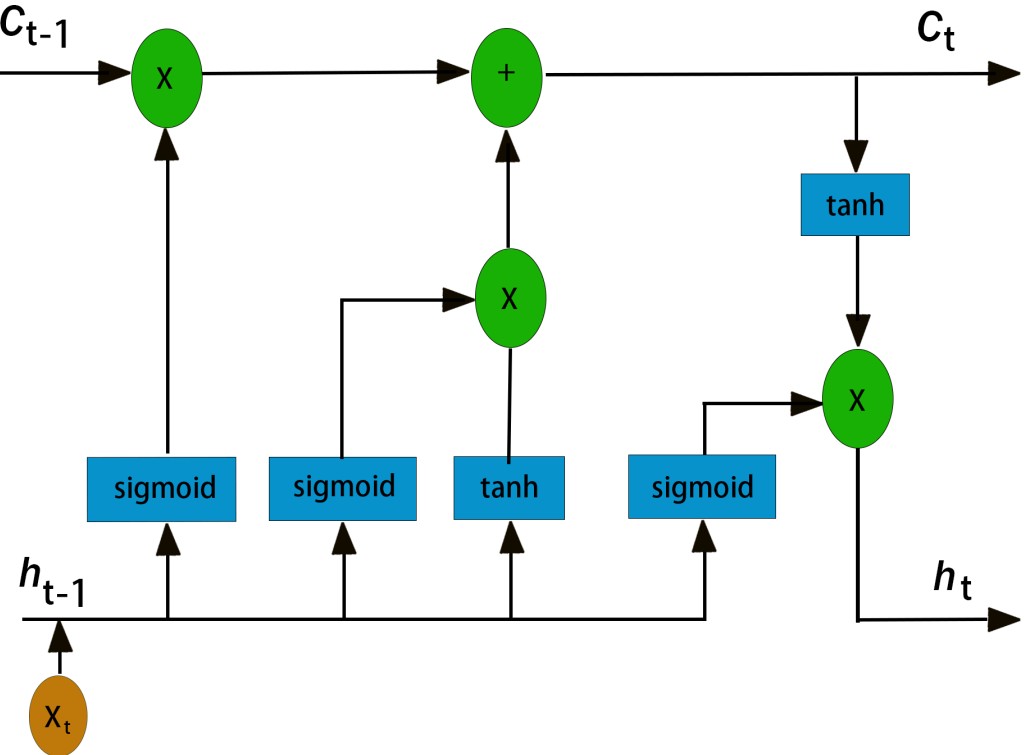

**Figure 4  LSTM network structure diagram.**

LSTM output. Finally, the prediction results of $PM_{2.5}$ are obtained by using the proposed model. We used the sigmoid function as the activation function (described in 'Activation function') instead of other commonly used activation functions. Because of its special structure, sigmoid function can alleviate the problem of gradient disappearance in neural networks to a certain extent. In addition, an efficient parameter optimizer, the adaptive moment estimation (Adam) optimizer, was used to replace the gradient descent method. In the Adam optimizer, the learning rate of the parameter can be dynamically updated. As a result, the model has more chances to escape local optima (*Zhang et al., 2021*).

## Setting model parameters

The root mean square error (RMSE), mean absolute percentage error (MAPE) and accuracy rate were used as the experimental evaluation indexes of the $PM_{2.5}$ prediction model. In the training phase of the neural network model, we first adopt a large learning rate for rapid adjustment to quickly reduce losses. We then use a decaying learning rate to further optimize the training of the network, eventually setting the learning rate to 0.001, the dropout rate to 0.5, the number of iterations to 200, the batch size to 32, and the time step to 4.In addition, by adjusting the experimental parameters and optimizing the number of layers, the optimal model structure parameters were selected, and the number of network layers in the CNN-LSTM model was set to 3 and the number of neurons contained in each hidden layer was set to 16. The specific experimental parameters are shown in Table 2. The
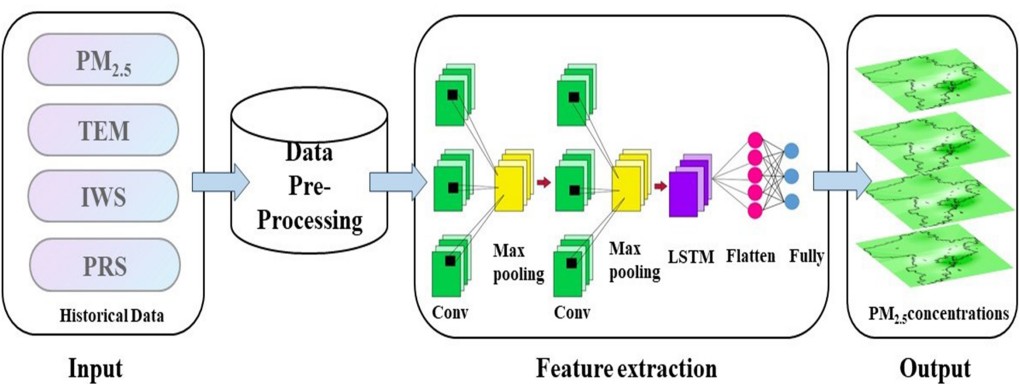

| Input | Feature extraction | Output |

**Figure 5 CNN-LSTM model technical flowchart.** Map created using ArcMap.

**Table 2 Model specific parameter settings.**

| Parameter | Value |
| --- | --- |
| Learning rate | 0.001 |
| Dropout rate | 0.5 |
| Epochs | 200 |
| Batch size | 32 |
| Time step | 4 |
| Number of layers of the model network | 3 |
| Depth of the model network | 16 |

Adam optimizer was used as the model optimizer. The L2 regularization term of the matrix composed of all variables in the model was added to the loss function to avoid overfitting (*Ryu & Park, 2022*).

## Activation function

The sigmoid activation function maps real numbers to the interval (0,1) and is mathematically defined as:

$$f(x) = \frac{1}{1 + e^{-x}}. \tag{8}$$

In neural networks, the sigmoid function is often used as the activation function of the output layer in binary classification problems, and its output can represent the probability that the sample belongs to the positive class. However, the sigmoid function also has some problems, such as the fact that the gradient of the sigmoid function is close to zero when it is far from zero, which may cause the gradient to disappear.

## Regularization

Model regularization has always been a core problem in machine learning, mainly with respect to improving the model generalizability. However, most models under the deep learning framework generally have many parameters. Thus, overfitting easily occurs. Therefore, many scholars have proposed regularization strategies suitable for deep learning

models, including dropout, which is a universal, computation-friendly but powerful regularization strategy. The basic principle of dropout is that during one iteration of the training model, some hidden layer neurons in the network are randomly discarded. These neurons can be considered as not a part of the network for the time being. However, their weights are retained, and they may be included in the next training iteration. The random exit mechanism of neurons can effectively prevent the interdependence of features.

**Evaluation index**

To evaluate the effect of the model, the coefficient of determination ($R^2$), RMSE and MAPE were used to compare the accuracy of the model. Among them, the RMSE and MAPE measure the deviation between the predicted value of model and the true value. The lower the values of these indicators are, the better the effect of the model. The $R^2$ reflects the degree of deviation between the predicted value of the model and the true value. The higher the $R^2$ value is, the lower the degree of deviation, and the better the effect of the model (*Liu et al., 2021*). The expressions of these three evaluation indicators are as follows:

$$RMSE = \sqrt{\frac{1}{n}\sum_{i=1}^{n}(y_i - \hat{y}_i)^2} \tag{9}$$

$$MAPE = \sum_{i=1}^{n}|\frac{y_i - \hat{y}_i}{y_i}| \cdot \frac{100}{n} \tag{10}$$

$$R^2 = 1 - \frac{\sum_{i=1}^{n}(y_i - \hat{y}_i)^2}{\sum_{i=1}^{n}(y_i - \bar{y})^2} \tag{11}$$

where n is the number of samples, $y_i$ is the i-th actual observation value, $\hat{y}_i$ is the corresponding model prediction value, and $\bar{y}_i$ is the mean of the actual observation values.

# EXPERIMENTAL SETUP AND RESULT ANALYSIS

## Data visualization analysis

To better understand the distribution characteristics of the $PM_{2.5}$ concentration in Qingdao, we present the average $PM_{2.5}$ concentration of 20 stations in Qingdao from January to December 2020 through a three-dimensional map. The three-dimensional map of the average $PM_{2.5}$ concentration of stations in January is shown in Fig. 6. The three-dimensional map of the average $PM_{2.5}$ concentration at the stations from February to December is shown in Fig. S6. Through the longitude and latitude of the 20 stations and the average concentration of $PM_{2.5}$, we can clearly see that the concentration of $PM_{2.5}$ in Qingdao is distributed in the range 30–70 μg/m$^3$. Due to coal burning in winter, the average $PM_{2.5}$ concentration at some stations in Qingdao in January was high, while only

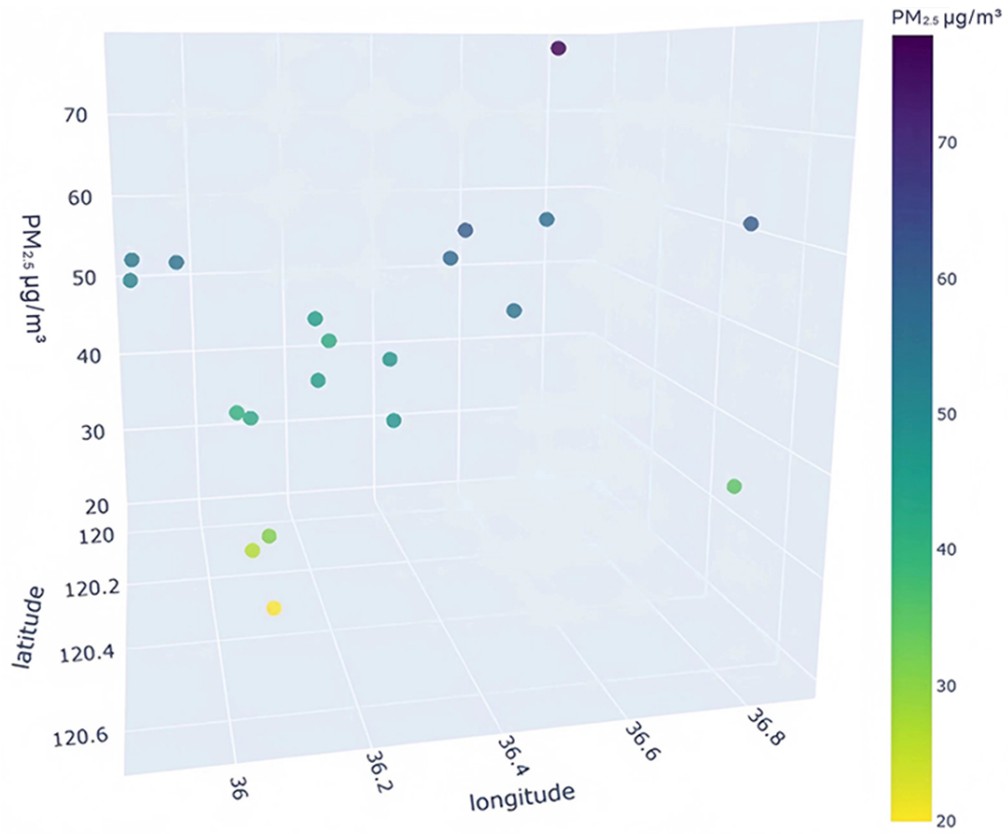

**Figure 6    Three-dimensional map of the average PM$_{2.5}$ concentration in January.**

a few stations had a low concentration. From this figure, we also note the necessity of predicting and controlling PM$_{2.5}$ concentrations in this region.

## Results of principal component analysis

We used the principal component analysis (PCA) method to rank the feature importance of factors affecting PM$_{2.5}$, laying a foundation for the subsequent input and analysis of the model parameters. In the process of preliminary selection of influencing factors, according to our field investigation and previous studies, compared with other factors, it is found that meteorological factors such as temperature, air pressure and wind speed are the main factors affecting atmospheric PM$_{2.5}$ in Qingdao and other coastal cities. Therefore, we ranked the influence of temperature, air pressure and wind speed on PM$_{2.5}$, as shown in Fig. 7. We know that the meteorological factors that have the greatest impact on PM$_{2.5}$ in Qingdao are pressure, followed by wind speed and temperature. Their proportions are 0.39, 0.38 and 0.23, respectively. Through the dimensionality reduction of the principal components, the original features were transformed into new principal components. We selected these three principal components and calculated the weight of the principal components by using the load coefficient, feature root and contribution rate information. The load coefficient was divided by the square root of the corresponding characteristic

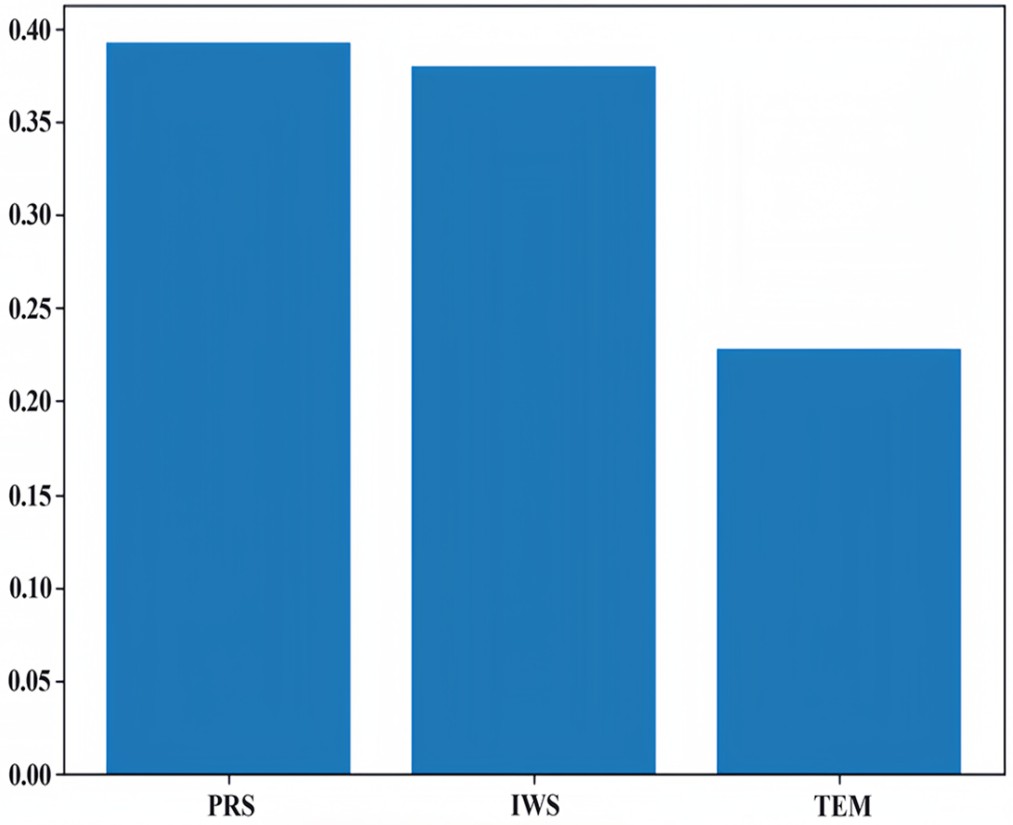

**Figure 7  Feature importance ranking.**

root to calculate the linear combination coefficient matrix, and the linear combination coefficient was multiplied by the contribution rate and then accumulated by the cumulative contribution rate to calculate the comprehensive score coefficient. Then, the weights of the comprehensive score coefficient of each factor in the total comprehensive score coefficient were −0.53, 0.62, and 0.58, respectively. The weight coefficients were combined with the original feature data as a new input variables for model prediction. A heatmap of the factor load matrix was plotted to show the strength of the linear relationship between each principal component and the original feature. A heatmap of the factor load matrix, as shown in Fig. 8, was plotted to show the strength of the linear relationship between each principal component and the original feature. From the figure, we note that the correlation of pressure to the original temperature is 0.94.

In summary, we can use the three principal components to capture the pressure, temperature and wind speed data in Qingdao. This information was used to improve the prediction model by calculating the weights of the principal components. This method may help to improve the accuracy and generalizability of the model to better predict the $PM_{2.5}$ concentration.

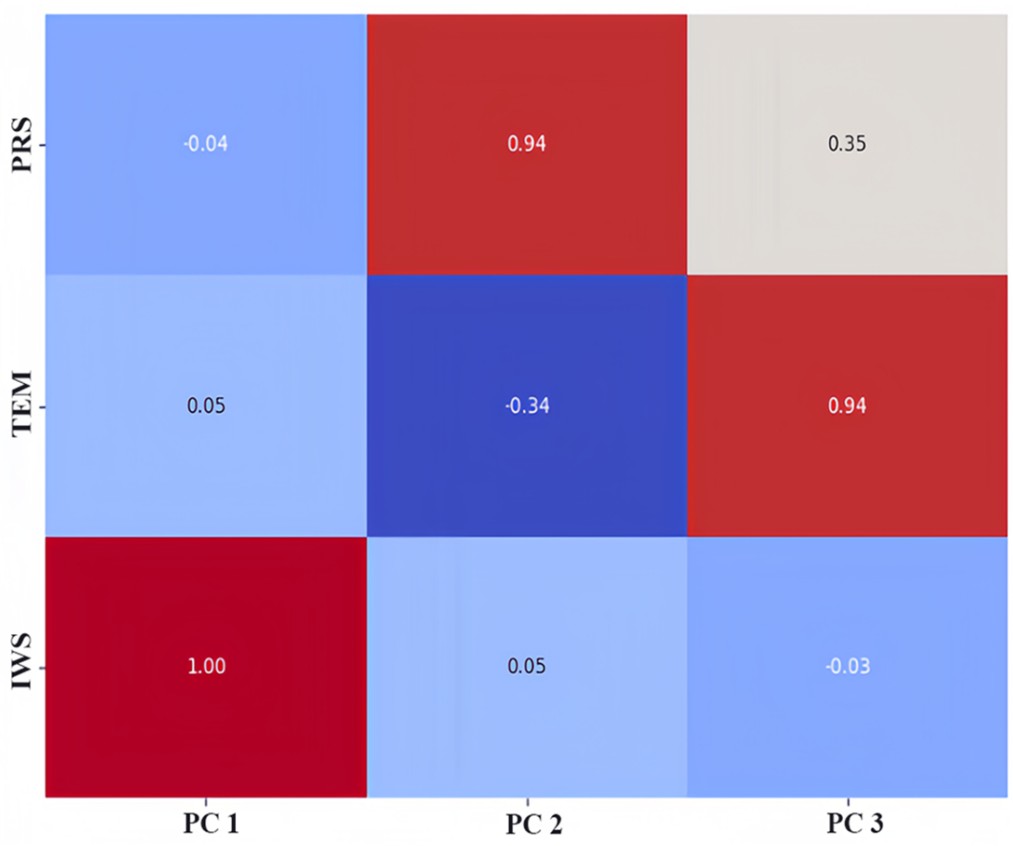

**Figure 8** Heatmap of the factor load matrix.

## Model performance evaluation results

To compare the performance of different models, we chose two commonly used neural networks, LSTM and the CNN. The prediction performance of different evaluation indicators is shown in Table 3, and the RMSE prediction results of each model are shown in Fig. 9. The MAPE prediction results are shown in Fig. 10. The fitted scatter density plot is shown in Fig. 11. It is not difficult to see that the prediction performance of our CNN-LSTM hybrid model is better than that of a single model. In particular, the predicted value of the CNN-LSTM model proposed in this article is in good agreement with the measured $PM_{2.5}$ value in Qingdao. In Table 3, the CNN-LSTM model achieves the lowest RMSE and MAPE values and the highest $R^2$ values in the daily forecast of air pollutants. By observing the RMSE and MAPE image curves of the three models in Fig. 9 and Fig. 10, it is evident that the difference between the predicted curve and the actual curve of the CNN-LSTM model is the smallest, followed by the CNN model, and the change trend between the predicted curve and the actual curve of the LSTM model is relatively large. This indicates that the accuracy of the proposed CNN-LSTM model is high, which means that the model can better capture the change and trend of the actual curve during the prediction process and can predict the future value or trend more accurately. The performance indicators of

**Table 3  Comparison of the prediction performance of different evaluation indices.**

| Model | RMSE | MAPE (%) | $R^2$ |
|---|---|---|---|
| CNN | 11.356 | 85.36 | 0.85 |
| LSTM | 14.367 | 54.21 | 0.83 |
| CNN-LSTM | 8.216 | 38.75 | 0.91 |

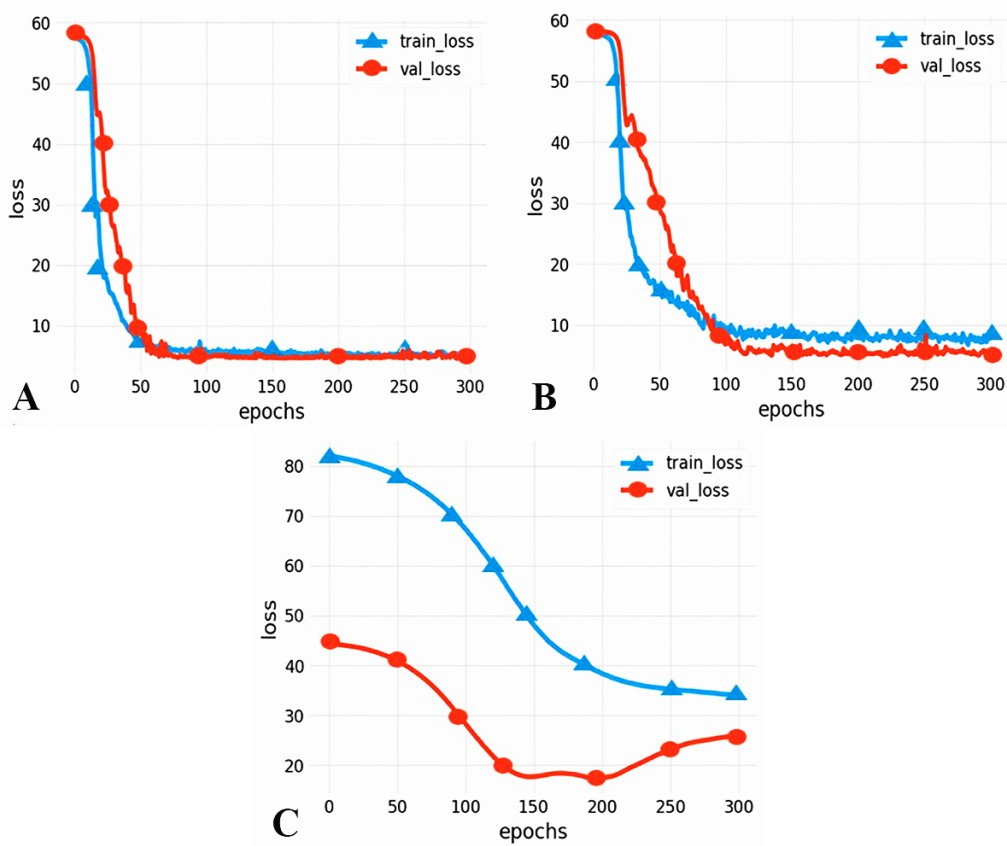

**Figure 9  RMSE prediction results of each model.** (A) CNN-LSTM, (B) CNN, (C) LSTM.

LSTM are as follows: RMSE of 14.367, MAPE of 54.21% and $R^2$ of 83%, while those of the CNN are as follows: RMSE of 11.356, MAPE of 85.36% and $R^2$ of 85%. The results show that the prediction accuracy of the deep neural network models including the CNN and CNN-LSTM is better than that of the LSTM model.

In general, from the experimental results, the prediction accuracy of the above two single models is lower than that of the CNN-LSTM model. In contrast, the CNN-LSTM model we constructed fully exploits the advantages of the two models, considers the spatiotemporal diffusion and correlation of the $PM_{2.5}$ concentration well, and reduces the prediction error. Therefore, the CNN-LSTM hybrid model has better prediction accuracy compared with that of the neural network.

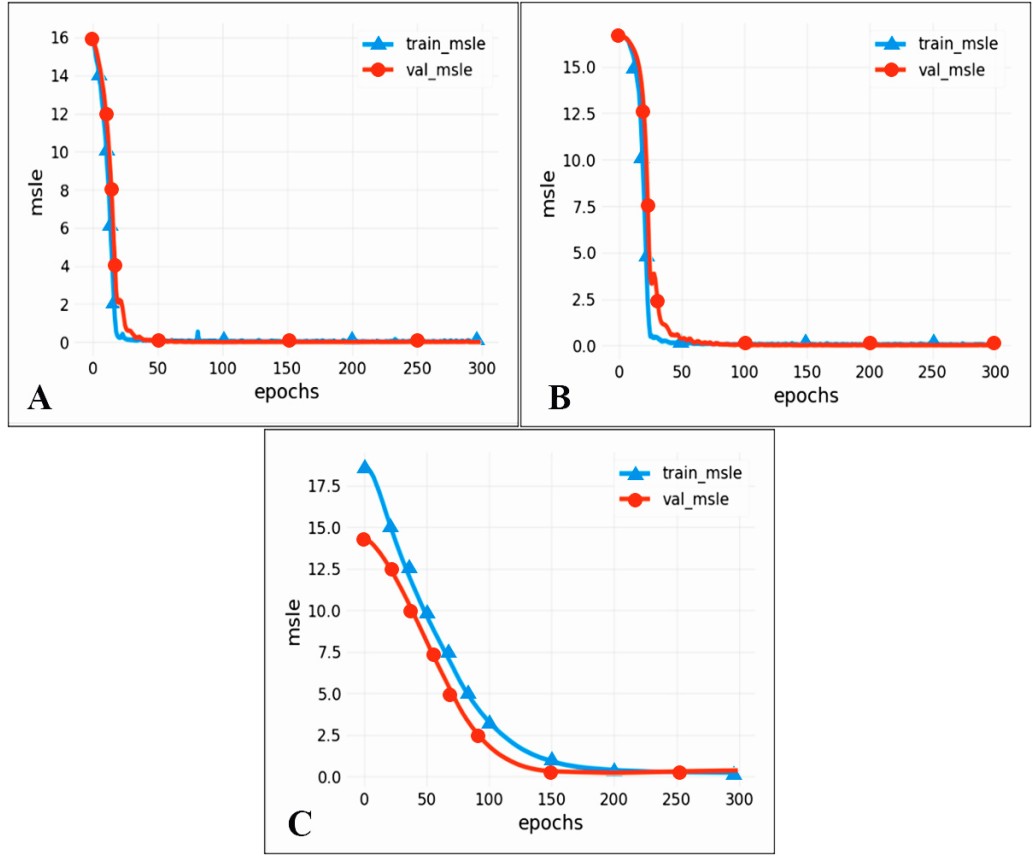

**Figure 10 MAPE prediction results.** (A) CNN-LSTM, (B) CNN, (C) LSTM.

## Analysis of the predicted performance results

In this study, a CNN-LSTM hybrid model based on temporal and spatial correlations was constructed to predict the average daily concentration of $PM_{2.5}$ at 20 stations in Qingdao from January 1, 2020, to December 31, 2020. The distribution of $PM_{2.5}$ stations in Qingdao is not uniform. In addition, the $PM_{2.5}$ concentrations at these 20 stations show a spatial trend of higher concentration in the north and lower concentration in the south, and the seasonal difference is very significant, with a pattern of winter > autumn > spring > summer. In addition, the improvement rate of the prediction effect for $PM_{2.5}$ sites in suburban areas is slightly higher than that in urban areas. This is mainly due to the relatively simple terrain and land type of the suburbs, as well as the small population density, so that the $PM_{2.5}$ emission sources are less than that of the cities, which is conducive to the establishment of spatial models (*Keyu, 2019*). In contrast, due to the high population density in urban areas, the interference of $PM_{2.5}$ emission sources and the influence of buildings on meteorological conditions, the accuracy of spatial information modeling is affected (*Moursi et al., 2022*). The predicted performance is shown in Fig. 12. The predicted values are close to the measured values over the whole region. The model has high accuracy, especially at sites with local high values, demonstrating that the CNN-LSTM hybrid algorithm based on

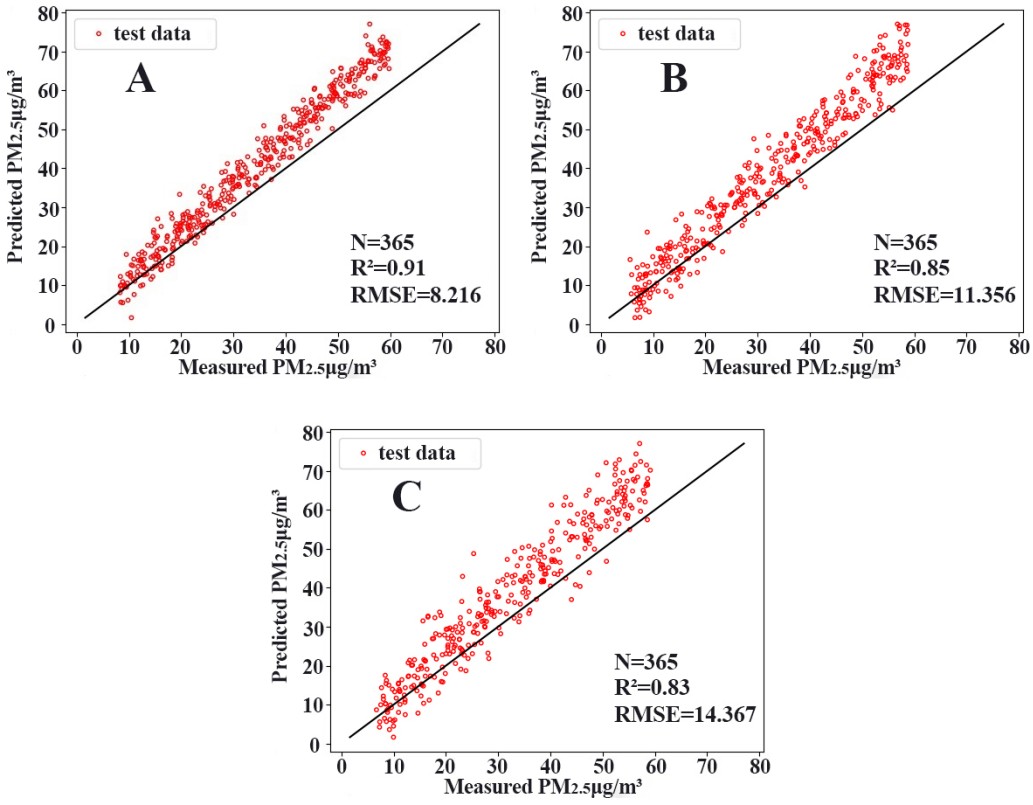

**Figure 11 Scatterplots.** (A) CNN-LSTM, (B) CNN, (C) LSTM.

temporal and spatial correlations can deal with nonlinear features and time series changes well. In particular, the performance index of the model shows that the model achieves high prediction accuracy and avoids the model overfitting problem, effectively proving that the CNN can effectively extract inherent features and thus improve the LSTM prediction. Therefore, the CNN-LSTM integrated algorithm has a good PM$_{2.5}$ prediction effect in Qingdao.

## RESULTS AND DISCUSSION

At present, many scholars have built different types of models to improve the accuracy of PM$_{2.5}$ estimation. For example, *Kang et al. (2021)* constructed an extreme gradient boosting (XGBoost) model, whose R$^2$ was 0.81; *Li et al. (2017)* proposed a deep learning method for geographic intelligence, and the site-based 10-fold cross-validation R$^2$ of the model was 0.82. *Zhenhong et al. (2020)* constructed a geographical neural network weighted regression (GNNWR) model, and the R$^2$ of the model was 0.83. In this article, the CNN-LSTM combined model was constructed using meteorological data and historical PM$_{2.5}$ concentration data. Qingdao City was selected as the study area, and daily PM$_{2.5}$ concentration prediction was carried out. The RMSE, MAPE and R$^2$ values were 8.216, 38.75% and 0.91, respectively, which were smaller than those of the single LSTM and

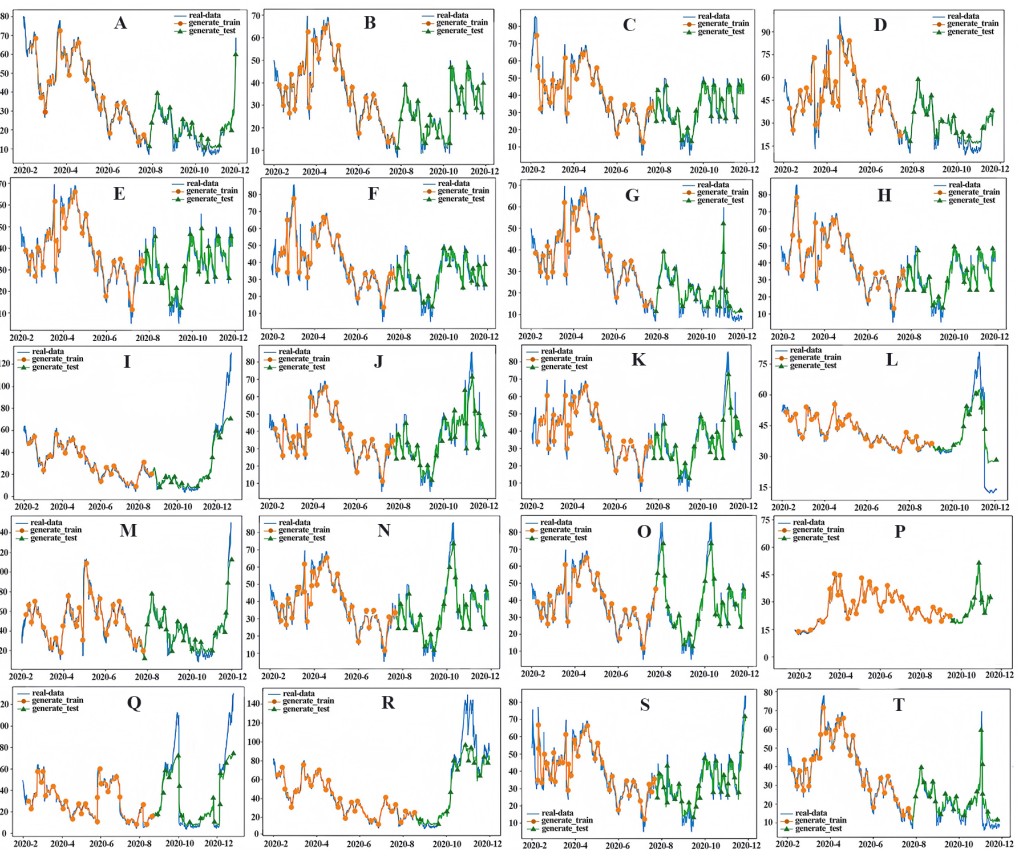

**Figure 12 PM$_{2.5}$ prediction chart.** (A) Yangkou, (B) Licang station, (C) Shibei substation, (D) Shinan east station, (E) Huangdao substation, (F) Sifang substation, (G) Shinan west, (H) Lacey, (I) Chengyang substation, (J) Huangdao district, (K) Develop substation, (L) Jiaonan station, (M) Jiaozhou station, (N) Jiaozhou, (O) Square substation, (P) Laoshan substation, (Q) Pingdu1, (R) Pingdu2, (S) Jimo1, (T) Jimo2).

CNN models. We used the properties of convolutional layers in convolutional neural networks and the convolution operation to extract important features from time series. Then, the processed data were sent to the LSTM network as input for training, and finally, the prediction result of the test set was obtained (*Qi et al., 2019*). This model not only considers the time series characteristics of the data but also accounts for the influence of other pollutant concentrations and meteorological factors on PM$_{2.5}$ concentration. Compared with a single LSTM model, the CNN-LSTM model has advantages in forecasting errors.

## CONCLUSION

(1) In view of the inability of existing machine models to capture the spatiotemporal relationship of PM$_{2.5}$, we constructed a CNN-LSTM neural network model. Given the complex nonlinear modeling capability of this model, the input factors were dynamically corrected. The model results indicated that the proposed model has

a higher estimation performance than that of the CNN and LSTM. The RMSE of CNN-LSTM was 8.216, the MAPE was 38.75%, and the $R^2$ was 0.91.

(2) In terms input factor selection, in addition to the $PM_{2.5}$ concentrations at Qingdao stations, meteorological factors such as pressure, temperature and wind speed were added. The experimental results showed that the addition of these meteorological factors can correct the estimation performance of $PM_{2.5}$ to a certain extent, making the spatial distribution of the estimation results more accurate.

(3) By using the proposed model to estimate the $PM_{2.5}$ concentration in Qingdao in 2020, it was found that the spatial trend of $PM_{2.5}$ is higher in the north than in the south, and the seasonal difference is also very significant, with a pattern of winter > autumn > spring > summer. There is much room for improvement of $PM_{2.5}$ in Qingdao, and more attention to this issue is needed.

(4) The model proposed in this study can better estimate the daily $PM_{2.5}$ concentration in Qingdao, and the estimated result was in good agreement with the ground observations. At the same time, the distribution of $PM_{2.5}$ stations in Qingdao is not uniform. However, the results of this study can compensate for the shortcomings of ground observations in spatial distribution. It should be noted the cofactors of the current model are mainly meteorological, and there is no such factor as an anthropogenic activity emission inventory with temporal and spatial characteristics. It is expected that such factors will be added to the model in the future, and more ground observation data will be accumulated for model training to obtain more accurate temporal and spatial distributions of $PM_{2.5}$ concentrations and provide more reliable data and theoretical support for China's ecological civilization construction.

## ACKNOWLEDGEMENTS

We are especially grateful to the editors, anonymous reviewers for appraising our manuscript and instructive comments, and Meteorological Bureau and Institute of Environmental Sciences of Shandong offering data.

### Funding

This work was supported by the Youth Foundation of Shandong Natural Science (ZR2023QD070), the Foundation of Chinese Academy of Sciences (B2-2023-0239), and the Key Research and Development Program for Shandong (2023RZA02017). The funders had no role in study design, data collection and analysis, decision to publish, or preparation of the manuscript.

### Grant Disclosures

The following grant information was disclosed by the authors:
The Youth Foundation of Shandong Natural Science: ZR2023QD070.
The Foundation of Chinese Academy of Sciences: B2-2023-0239.
Key Research and Development Program for Shandong: 2023RZA02017.

## Competing Interests

The authors declare there are no competing interests.

## Author Contributions

- Xuesong Bai conceived and designed the experiments, performed the experiments, analyzed the data, prepared figures and/or tables, authored or reviewed drafts of the article, and approved the final draft.
- Na Zhang performed the experiments, prepared figures and/or tables, and approved the final draft.
- Xiaoyi Cao performed the experiments, authored or reviewed drafts of the article, and approved the final draft.
- Wenqian Chen analyzed the data, authored or reviewed drafts of the article, and approved the final draft.

## Data Availability

The raw data are available in the Supplemental Files.

## Supplemental Information

Supplemental information for this article can be found online at http://dx.doi.org/10.7717/peerj.17811#supplemental-information.

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
