# Peer review of "Prediction of PM2.5 concentration based on a CNN-LSTM neural network algorithm"

_PeerJ, doi:10.7717/peerj.17811_

## Round 0.1 · original submission · Minor Revisions

This manuscript can be considered for publication once these issues have been addressed properly.

Reviewer 1 ·

Basic reporting

The CNN-LSTM model proposed in the paper performs better than CNN and LSTM, with an RMSE of 8.216, MAPE of 38.75%, and R2 of 0.91. It has higher estimation performance, captures the spatiotemporal relationship of station PM2.5 well, uses reasonable technical methods, and has certain research value. It is recommended to minor revision and accept it.

Experimental design

A convolutional neural network Long short-term memory (CNN-LSTM) model was constructed. By optimizing the model structure and algorithm, the spatial-temporal characteristics and learning time correlation were effectively extracted, and the spatial correlation between sites was processed to improve the accuracy of regional PM2.5 prediction. The experimental parameter design of the model is listed and the performance of the model is evaluated. The site PM2.5 was predicted and the performance of the integrated CNN-LSTM algorithm was compared with that of CNN and LSTM to further verify its effectiveness.

Validity of the findings

The data source of the experiment is reliable, and the parameter settings of the optimization experiment are optimized. The effectiveness of parameter settings was verified through training and evaluation of the model, ensuring the reproducibility and comparability of the experiment. A comprehensive evaluation was conducted on the performance of the CNN-LSTM integrated model in PM2.5 concentration prediction tasks. The final results indicate that the CNN-LSTM model has high prediction accuracy and stability. This indicates the potential application prospects of the CNN-LSTM integrated algorithm in the field of air quality prediction, providing a reference basis for further research and practice.

Additional comments

1、Abstract: The purpose of this paper is to predict and analyze the PM2.5 concentration of the station based on convolutional neural network Long short-term memory (CNN-LSTM) model with comprehensive deep learning. The PM2.5 data of Qingdao in 2020 were collected for pre-processing and characteristic analysis. Whether only the PM2.5 data of the site is used.
2、Table 1 Some fonts in this picture are not very clear, which affects reading.
3、The sigmoid function is used as the activation function because the sigmoid function can solve the problem of gradient disappearance in neural networks due to its special structure. The word "solution" is not used accurately.
4、In the model parameter setting, the number of network layers is 3 and the number of neurons is 16, so the prediction effect of PM2.5 is the best. Does this 3 refer to the depth of the overall model, or to the depth of the LSTM alone?

Reviewer 2 ·

Basic reporting

The language should be checked thoroughly because many mistakes, such as subscript in the manuscript and figures (Line 171-173, Figure6-8), repeated information (Line 112), excessive words (Line 260-261).

Experimental design

No comments

Validity of the findings

The parameters for evaluating the prediction effects seems good. How about the advantage compared with the other previous studies, such as mentioned in Line 341-345?

Additional comments

Based on deep learning technology, this study improved and constructed the convolutional neural network short Term memory network (CNN-LSTM) model, integrated intelligent deep learning to predict and analyze the change characteristics of PM2.5 concentration at stations. The research method and data source are reliable, the research objective is clear, and the analysis of the paper is reasonable. It is suggested to receive the paper after modification.
1、 The article only introduced that the meteorological factors were obtained from the China Meteorological Data Service Center, but did not elaborate on how the meteorological factors were processed.
2、 Redevelop the overview map of the research area, which should not contain Chinese fonts. Modify the format of the latitude and longitude coordinates.
3、 In section 2.1, air pollution will become one of the factors restricting sustainable development. whether the "will" here is inaccurate.
4、 In order to input the output of the CNN into the LSTM, a flat layer is constructed between the CNN and the LSTM, which does not correspond to the model diagram.
5、 A table may deliver the information on Line 219-222 more clearly.
6、 The title and number of the Figures should be checked and corrected. There are no differences between Figure 9, 10, 11, and 12. Each sub-figure should be numbered.
7、 There are many supporting figures. But they were not mentioned in the manuscript.

Annotated reviews are not available for download in order to protect the identity of reviewers who chose to remain anonymous.

Reviewer 3 ·

Basic reporting

In this study, Qingdao city is taken as the research object, and the CNN-LSTM deep learning model is used to predict PM2.5 concentration by combining PM2.5 concentration and meteorological data.Through the extraction of spatio-temporal features and the learning of time series dependence, the model shows high accuracy and stability in PM2.5 concentration prediction.The results showed that the model had an R2 of 0.91 and an RMSE of 8.216 μg/m3, which were better than the prediction results of CNN or LSTM model alone.In addition, the study also found that meteorological factors such as air temperature, air pressure and wind speed have significant effects on PM2.5 concentration in Qingdao. Overall, this study improves the prediction performance of PM2.5 concentration and reveals the association between PM2.5 concentration and meteorological factors through the deep learning method, which is of great significance to improve the quality of the atmospheric environment and protect public health.
Firstly, this study discusses the existing research, then identifies the shortcomings of the existing research, and proposes the research content and the scientific problems to be solved in this study in response to the shortcomings. Therefore, the structure of the article is scientific and reasonable, the article expresses accurate views, the use of correct words, smooth sentences, part and part of the connection between the closer. However, some of the problems of the article need to be further improved and improved, and the author is invited to see the deficiencies and suggestions part of the paper for specific comments.

Experimental design

no comment

Validity of the findings

no comment

Additional comments

(139 lines) deleting the data with continuous null values, filling the remaining small amount of missing data with previous data, modifying the format of the dataset so that it can be read by the neural network 1.The basis for data imputation (specific methods) includes?
(274 lines)
2.Is it sufficient to select only three meteorological factors (input factor in the text) as characterisation variables in the results of the principal component analysis? And there is no explanation for the selection of these categories (although the authors mention in the data source that PM2.5 imposes a strong random effect in space and time, it is insufficient to select only these categories).In the research status section of the article, it is mentioned that other scholars at home and abroad have selected a variety of feature variables in their previous studies, but the results of this paper mainly discuss the accuracy of the CNN-LSTM model with only three feature variables selected after fusion is higher than that of a single model (R2= 0.91, RMSE=8.216 μg/m3);If the authors performed the step of optimal selection of the characteristic variables (input factor), it is recommended that a specific description of this step be included in the article, along with the rationale for why the three meteorological factors of barometric pressure, wind speed, and temperature were selected.
(328、329 lines)
3.The paper mentions in sentences 328, 329 that overly simple topography and overly homogeneous land use types are favourable for spatial modelling (this is not only related to these two points).The author can argue this point from other sources, or bring in other people's papers to support it, and the description of urban areas in the text needs to be revised in the same way.
4.The study period is one year in 2020, how many valid data can be matched after temporal and spatial matching of the data? Can the dataset used for the study meet the requirements of a large training dataset for deep learning?
5.The reference to extreme wind speed (EWS, m/s) in 2.2 is incorrectly abbreviated in Figures 7 and 8; it is recommended that Study Area Profile Figure 1 be recreated and that the legend be presented in English.

---

## Round 0.2 · Minor Revisions

I have taken responsibility for your submission as academic editor and can confirm that your revision has solved most of the issues, including the linguistic deficits. However, there are still a few issues identified by reviewer 2 that should be resolved before final acceptance of the manuscript. I therefore request a revision in the short term and look forward to your submission of the revised manuscript.

Reviewer 2 ·

Basic reporting

Most errors and suggestions have been revised. It is suggested to directly receive the paper after the minor modification.
1、 The unit should be checked again. For example, ug/m3 in the manuscript should be µg/m3.
2、 The first letter of “station” in Table 1 should be capitalized. And similar mistakes should be corrected in Table 2.
3、 Please check the Figures, such as the title in the vertical coordinate in Figure 6 and the units in all figures.
4、 Figure 12-16 are all PM2.5 prediction charts, their titles are same, why draw them separately? It is better to dram them together.

Experimental design

Based on deep learning technology, this study improved and constructed the convolutional neural network short Term memory network (CNN-LSTM) model, integrated intelligent deep learning to predict and analyze the change characteristics of PM2.5 concentration at stations. The Experimental design is reliable.

Validity of the findings

The analysis of the paper is reasonable and the findings are reliable.

---

## Round 0.3 · accepted · Accept

Thank you for the revision of the manuscript. I hereby certify that you have adequately taken into account the reviewer's comments and improved the manuscript accordingly. Based on my assessment as an Academic Editor, your manuscript is now ready for publication.